# Physiological and metabolic insights into the first cultured anaerobic representative of deep-sea *Planctomycetes* bacteria

**Rikuan Zheng[1,2,3], Chong Wang[1,2,3], Rui Liu[1,2,3], Ruining Cai[1,2,3,4], Chaomin Sun[1,2,3,4]***

[1]CAS and Shandong Province Key Laboratory of Experimental Marine Biology & Center of Deep Sea Research, Institute of Oceanology, Chinese Academy of Sciences, Qingdao, China; [2]Laboratory for Marine Biology and Biotechnology, Qingdao National Laboratory for Marine Science and Technology, Qingdao, China; [3]Center of Ocean Mega-Science, Chinese Academy of Sciences, Qingdao, China; [4]College of Earth Science, University of Chinese Academy of Sciences, Beijing, China

**\*For correspondence:**
sunchaomin@qdio.ac.cn

**Competing interest:** The authors declare that no competing interests exist.

**Abstract** *Planctomycetes* bacteria are ubiquitously distributed across various biospheres and play key roles in global element cycles. However, few deep-sea *Planctomycetes* members have been cultivated, limiting our understanding of *Planctomycetes* in the deep biosphere. Here, we have successfully cultured a novel strain of *Planctomycetes* (strain ZRK32) from a deep-sea cold seep sediment. Our genomic, physiological, and phylogenetic analyses indicate that strain ZRK32 is a novel species, which we propose be named: *Poriferisphaera heterotrophicis*. We show that strain ZRK32 replicates using a budding mode of division. Based on the combined results from growth assays and transcriptomic analyses, we found that rich nutrients, or supplementation with $NO_3^-$ or $NH_4^+$ promoted the growth of strain ZRK32 by facilitating energy production through the tricarboxylic acid cycle and the Embden-Meyerhof-Parnas glycolysis pathway. Moreover, supplementation with $NO_3^-$ or $NH_4^+$ induced strain ZRK32 to release a bacteriophage in a chronic manner, without host cell lysis. This bacteriophage then enabled strain ZRK32, and another marine bacterium that we studied, to metabolize nitrogen through the function of auxiliary metabolic genes. Overall, these findings expand our understanding of deep-sea *Planctomycetes* bacteria, while highlighting their ability to metabolize nitrogen when reprogrammed by chronic viruses.

## eLife assessment

This **important** study advances the understanding of physiological mechanisms in deep-sea Planctomycetes bacteria, revealing unique characteristics such as the only known Phycisphaerae using a budding mode of division, extensive involvement in nitrate assimilation, and release phage particles without cell death. The study uses **convincing** evidence based on experiments using growth assays, phylogenetics, transcriptomics, and gene expression data. The work will be of interest to bacteriologists and microbiologists in general.

## Introduction

*Planctomycetes* bacteria are ubiquitous in many environments, including lakes (*Pollet et al., 2011*), wetlands (*Dedysh and Ivanova, 2019*), soil (*Buckley et al., 2006*), freshwater (*Brümmer et al., 2004*), oceanic waters, and abyssal sediments (*Woebken et al., 2007*; *Goffredi and Orphan, 2010*), where

they are critical for carbon and nitrogen cycling, although the specific pathways used are unknown (*Wiegand et al., 2018*). Although *Planctomycetes* bacteria are highly abundant in nature, relatively few have been cultivated; therefore, the unexplored groups lack cultured and characterized representatives (*Fuerst and Sagulenko, 2011*). Currently, all taxonomically described *Planctomycetes* with validly published names can be divided into two recognized classes: Planctomycetia and Phycisphaerae (*Fukunaga et al., 2009*; *Wiegand et al., 2018*). *Candidatus* Brocadiae is considered a third class within the phylum Planctomycetes, although pure cultures are not yet available for its members and its class-level status remains to be defined (*Kartal et al., 2012*; *Kartal et al., 2013*). The majority of the species isolated from the phylum Planctomycetes are from the class, Planctomycetia — a class that consists of four orders: Planctomycetales, Pirellulales, Gemmatales, and Isosphaerales (*Dedysh et al., 2020*). In contrast, only few members from the Phycisphaerae class have been cultured; the class Phycisphaerae contains three orders: Phycisphaerales (*Fukunaga et al., 2009*), Tepidisphaerales (*Kovaleva et al., 2015*), and Sedimentisphaerales (*Spring et al., 2018*).

*Planctomycetes* bacteria have fascinating physiological characteristics (*Wiegand et al., 2018*). For decades, *Planctomycetes* bacteria have blurred the lines between prokaryotes and eukaryotes. *Planctomycetes* bacteria possess several uncommon traits when compared with typical bacteria: they have a compartmentalized cell plan, an enlarged periplasm, a tightly folded nucleus-like structure, an endocytosis-like method of uptake, and a FtsZ-free method of cell division (*Fuerst and Webb, 1991*; *Lindsay et al., 1997*; *Lonhienne et al., 2010*; *Wiegand et al., 2018*). The unique cellular structures in *Planctomycetes* bacteria have stretched and challenged our understanding of the concept of a 'prokaryote' (*Fuerst and Sagulenko, 2011*). *Planctomycetes* bacteria also exhibit a diverse range of respiration and cell division methods. Members of the class Planctomycetia are mostly aerobic and divide by budding, while Phycisphaerae members are mostly anaerobic and divide by binary fission (*Rivas-Marín et al., 2016*; *Wiegand et al., 2018*; *Pradel et al., 2020*; *Wiegand et al., 2020b*). However, a recent report showed that *Poriferisphaera corsica* KS4 — a novel strain from class Phycisphaerae — was aerobic and that it might divide by budding (*Kallscheuer et al., 2020*); this is different from previous reports and suggests that there are many physiological and cellular *Planctomycetes* bacteria characteristics yet to be discovered.

The deep sea is where life may have originated from and where stepwise evolution occurred (*Orgel, 1998*); it is also where a large number of uncultured microorganisms live (*Zheng et al., 2021a*). Among these microbes, *Planctomycetes* might dominate deep-sea sediments (*Wiegand et al., 2018*). For example, in the Gulf of Mexico, the phylum Planctomycetes accounts for 28% of all bacteria, where they seem to be involved in the nitrogen cycle, and the breakdown of organic detrital matter, which is delivered to the sediment as marine snow (*Vigneron et al., 2017*). Unfortunately, only few Planctomycetes bacteria from deep-sea environments have been cultured (*Wang et al., 2020*; *Wiegand et al., 2020b*), limiting our understanding of their characteristics (e.g. material metabolism, element cycling, and ecological role) (*Kulichevskaya et al., 2020*).

Here, we successfully cultured a novel member of *Planctomycetes* (strain ZRK32) from a deep-sea subsurface sediment. We found that strain ZRK32 used a budding mode of division. We also found that supplementation with rich nutrients, and either $NO_3^-$ or $NH_4^+$, promoted strain ZRK32 growth. Moreover, the presence of $NO_3^-$ or $NH_4^+$ induced strain ZRK32 to release a bacteriophage in a chronic manner — a process that does not kill the host cell (i.e. the host bacteria continues to grow despite phage reproduction) (*Liu et al., 2022*). This bacteriophage reprogrammed strain ZRK32, and another marine bacterium we studied, to metabolize nitrogen through the action of auxiliary metabolic genes (AMGs).

## Results

### Isolation, morphology, and phylogenetic analysis of a novel strain of *Planctomycetes* isolated from a deep-sea cold seep

We enriched deep-sea *Planctomycetes* bacteria using a basal medium supplemented with rifampicin and inorganic nitrogen sources ($NaNO_3$ and $NH_4Cl$). These enriched samples were plated on to agar slants in Hungate tubes, and then colonies with distinct morphologies were selected and cultivated (*Figure 1A*). Some colonies were noted to be from the phylum Planctomycetes, based on their 16S rRNA sequences. Among these, strain ZRK32 was selected for further study as it grew faster than

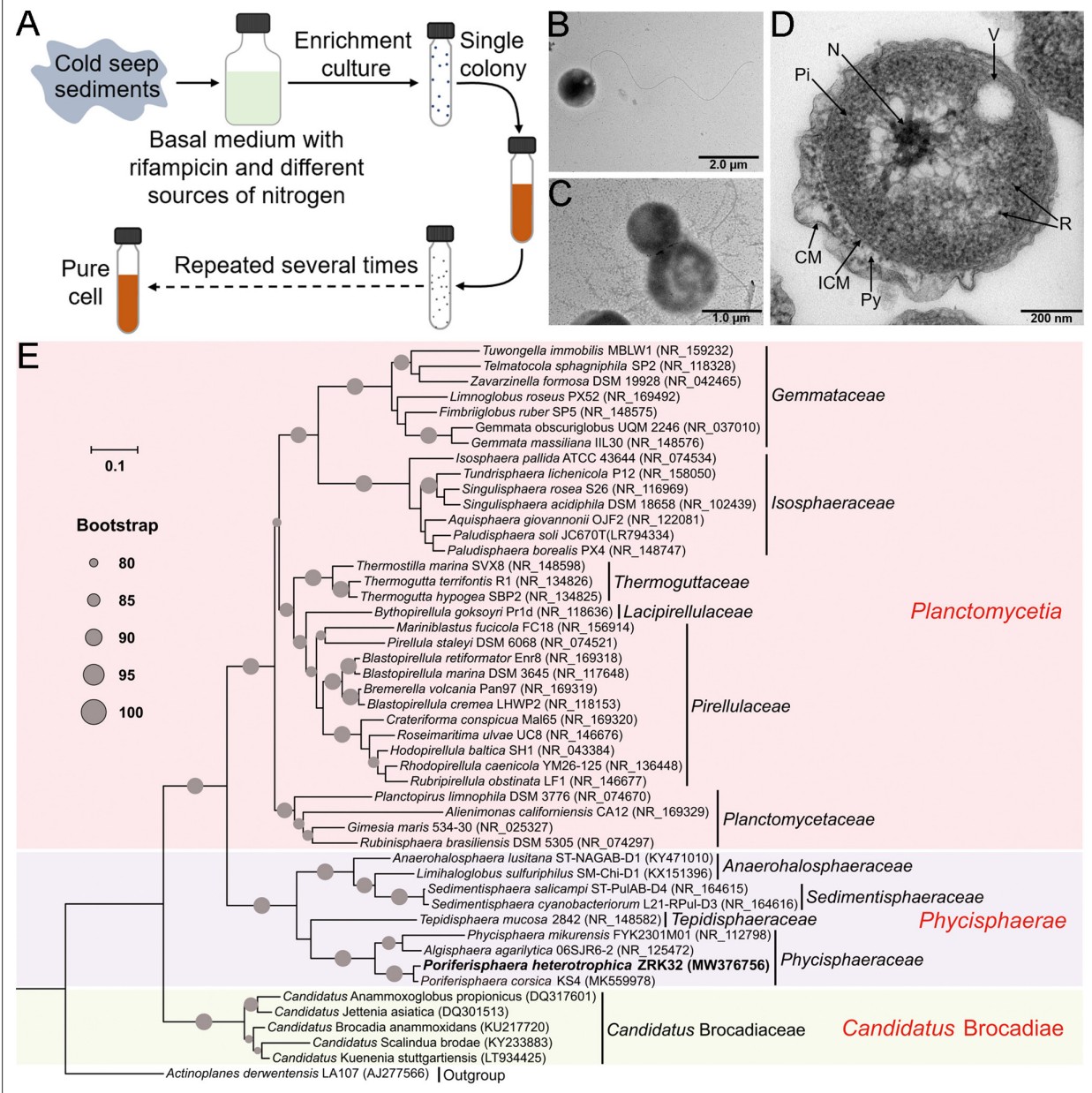

**Figure 1.** Isolation, morphology, and phylogenetic analysis of *Poriferisphaera heterotrophicis* ZRK32. (**A**) Diagram showing the strategy used to isolate the *Planctomycetes* bacteria. (**B, C**) Transmission electron microscope (TEM) observation of strain ZRK32. (**D**) TEM observation of ultrathin sections of cells from strain ZRK32. Abbreviations: CM, outer membrane; Pi, cytoplasm; R, ribosome; N, nucleoid; ICM, cytoplasmic membrane; Py, periplasm; V, vesicle-like organelles. (**E**) Phylogenetic analysis of strain ZRK32. Phylogenetic placement of strain ZRK32 within the phylum Planctomycetes, based on almost complete 16S rRNA gene sequences. The NCBI accession number for each 16S rRNA gene is indicated after each corresponding strain's name. The tree was inferred and reconstructed using the maximum likelihood criterion, with bootstrap values (%)>80; these are indicated at the base of each node with a gray dot (expressed as a percentage from 1000 replications). The 16S rRNA gene sequence of *Actinoplanes derwentensis* LA107[T] was used as the outgroup. Bar, 0.1 substitutions per nucleotide position.

The online version of this article includes the following figure supplement(s) for figure 1:

**Figure supplement 1.** Transmission electron microscope (TEM) observation of the morphology of cells from *P. heterotrophicis* ZRK32 (**A, B**).

**Figure supplement 2.** Maximum likelihood phylogenetic tree of genome sequences from the *P. heterotrophicis* ZRK32 and other *Planctomycetes* bacteria constructed from the concatenated alignment of 37 single-copy genes; *A. derwentensis* LA107 was used as the outgroup.

**Figure supplement 3.** Circular diagram of the *P. heterotrophicis* ZRK32 genome.

**Figure supplement 4.** Physiological characterizations of *P. heterotrophicis* ZRK32.

**Figure supplement 5.** Ultrathin transmission electron microscope (TEM) sections showing some eukaryote-like structures observed in cells from *P. heterotrophicis* ZRK32.

other strains. Following negative staining and transmission electron microscope (TEM) observation, we observed that strain ZRK32 cells were spherical (with an average diameter of 0.4–1.0 μm), and had a single polar flagellum (*Figure 1B*). Moreover, we found that the mother and daughter cells of strain ZRK32 had distinct sizes at the stage of cell division (*Figure 1C* and *Figure 1—figure supplement 1*), indicating that strain ZRK32 was dividing asymmetrically (i.e. through budding). Ultrathin whole-cell sections showed that strain ZRK32 possessed a condensed and intact nucleoid-like structure, and a complex extended membrane structure (*Figure 1D*).

Based on the 16S rRNA sequence of strain ZRK32, a sequence similarity calculation using the NCBI server indicated that the closest relatives of strain ZRK32 were *P. corsica* KS4[T] (98.06%), *Algisphaera agarilytica* 06SJR6-2[T] (88.04%), *Phycisphaera mikurensis* NBRC 102666[T] (85.28%), and *Tepidisphaera mucosa* 2842[T] (82.94%). Recently, the taxonomic threshold for species based on 16S rRNA gene sequence identity value was 98.65% (*Kim et al., 2014*). Based on these criteria, we proposed that strain ZRK32 might be a novel representative of the genus *Poriferisphaera*. In addition, to clarify the phylogenetic position of strain ZRK32, the genome relatedness values were calculated by the average nucleotide identity (ANI), the tetranucleotide signatures (Tetra), and in silico DNA-DNA hybridization (*is*DDH), against the genomes of strains ZRK32 and KS4. The ANIb (ANI using BLAST), ANIm (ANI using MUMmer), Tetra, and *is*DDH values were 72.89%, 85.34%, 0.97385, and 20.90%, respectively (*Supplementary file 1*). These results together demonstrated the strain ZRK32 genome to be obviously below established 'cut-off' values (ANIb: 95%, ANIm: 95%, Tetra: 0.99, *is*DDH: 70%) for defining bacterial species, suggesting strain ZRK32 represents a novel strain within the genus *Poriferisphaera*.

To further confirm the taxonomy of strain ZRK32, we performed phylogenetic analyses. The maximum likelihood tree of 16S rRNA indicated that strain ZRK32 was from the genus *Poriferisphaera* and that it formed an independent phyletic line with strain *P. corsica* KS4[T] (*Figure 1E*). The genome tree also suggested that this novel clade was a sister strain of strain KS4[T], which belongs to the genus *Poriferisphaera* (*Figure 1—figure supplement 2*). Based on its genomic (*Figure 1—figure supplement 3* and *Supplementary file 1*), physiological (*Figure 1—figure supplement 4*), and phylogenetic characteristics, strain ZRK32 was distinguishable from strain *P. corsica* KS4[T], which is currently the only species of the genus *Poriferisphaera* with a validly published name. We therefore propose that strain ZRK32 represents a novel species in the genus *Poriferisphaera*, for which the name *P. heterotrophicis* sp. nov. is proposed.

To understand more characteristics of strain ZRK32, its whole genome was sequenced and analyzed. The genome size of strain ZRK32 was 5,234,020 bp with a DNA G+C content of 46.28 mol% (*Figure 1—figure supplement 3*). Annotation of the genome of strain ZRK32 revealed that it consisted of 4175 predicted genes including 6 rRNA genes (2, 2, and 2 for 5S, 16S, and 23S, respectively) and 45 tRNA genes, which were higher than those reported in the most closely related type strain *P. corsica* KS4[T] (*Supplementary file 1*). Moreover, the genome size (5,234,020 bp) and gene numbers (4175) of strain ZRK32 were also higher than those in strain KS4[T] (4,291,168 bp, 3714). Strain ZRK32 was able to grow over a temperature range of 4–32°C (optimum, 28°C), which was wider than that of strain KS4[T] (15–30°C, optimum 27°C) (*Figure 1—figure supplement 4A*). The pH range for growth of strain ZRK32 was 6.0–8.0 (optimum, pH 7.0) (*Figure 1—figure supplement 4B*). Growth of strain ZRK32 was observed at 0.5–5.0% NaCl (*Figure 1—figure supplement 4C*).

## Description of *P. heterotrophicis* sp. nov

*P. heterotrophicis* (hetero'tro.phicis. L. fem. adj. *heterotrophicis* means a heterotrophic lifestyle). Cells are spherical, average diameter of 0.4–1.0 μm, strictly anaerobic, and have a single polar flagellum. The temperature range for growth is 4–32°C with an optimum at 28°C. Growing at pH values of 6.0–8.0 (optimum, pH 7.0). Growth occurs at NaCl concentrations from 0.5% to 5.0%. The type strain, ZRK32[T], was isolated from a deep-sea cold seep sediment, P.R. China. The DNA G+C content of the type strain is 46.28 mol%.

## Rich nutrients promote *P. heterotrophicis* ZRK32 growth

The growth rate of strain ZRK32 increased when it was cultured in a rich medium (containing 10 times more yeast extract than basal medium) (*Figure 2A*). To gain further insight into its metabolic characteristics, we performed transcriptomic analyses of strain ZRK32 grown in the rich medium and strain ZRK32 grown in the basal medium. The results showed that the expression of many genes

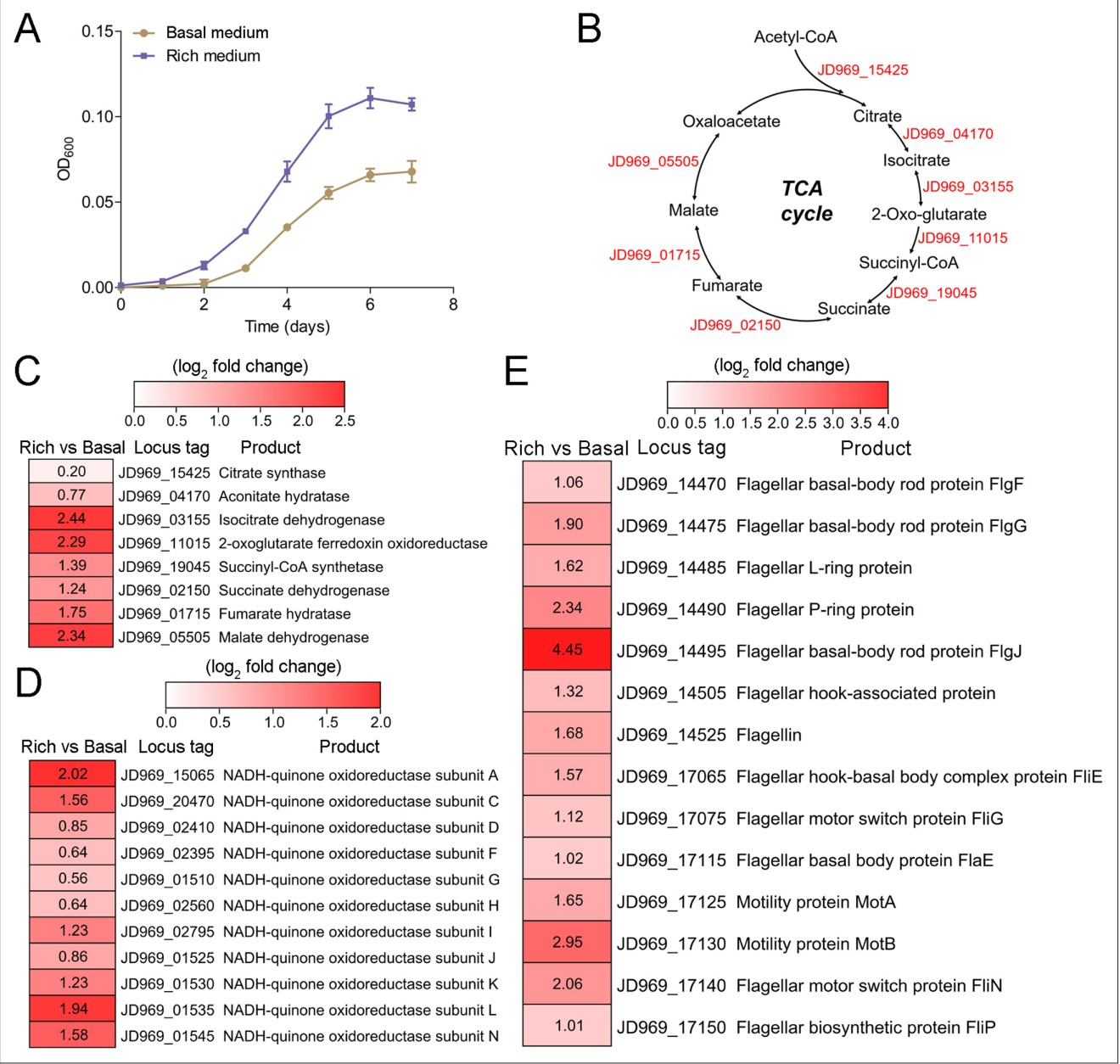

**Figure 2.** Growth assay and transcriptomic analysis of *P. heterotrophicis* ZRK32 strains cultivated in basal medium and rich medium. (**A**) Growth curves of ZRK32 strains cultivated in basal medium and rich medium. Data shown as mean; error bars = SD (Standard Deviation). (**B**) Diagram of the tricarboxylic acid (TCA) cycle. The gene numbers shown in this schematic are the same as those shown in panel C. Transcriptomics-based heat map showing the relative expression levels of genes associated with the TCA cycle (**C**), NADH-quinone oxidoreductase (**D**), and flagellar assembly (**E**) of strain ZRK32 cultivated in rich medium (Rich) compared with strain cultivated in basal medium (Basal). The numbers in panels C, D, and E represent the fold change of gene expression (by using the $\log_2$ value).

The online version of this article includes the following figure supplement(s) for figure 2:

**Figure supplement 1.** Transcriptomics analysis of the genes associated with the Embden-Meyerhof-Parnas (EMP) glycolysis pathway of *P. heterotrophicis* ZRK32 strains cultivated in the rich medium alone and cultivated in rich medium supplemented with either 20 mM $NO_3^-$, 20 mM $NH_4^+$, or 20 mM $NO_2^-$.

**Figure supplement 2.** Real-time quantitative reverse transcription PCR (qRT-PCR) detection of the relative expression levels of the genes associated with the tricarboxylic acid (TCA) cycle (**A**), NADH-quinone oxidoreductase (**B**), flagellar assembly (**C**), and Embden-Meyerhof-Parnas (EMP) glycolysis pathway (**D**) of *P. heterotrophicis* ZRK32 strains cultivated in rich medium (Rich) compared with strains cultivated in basal medium (Basal).

**Figure supplement 3.** Growth curves of ZRK32 strains cultivated in the basal medium, basal medium supplemented with 5 g/L *N*-acetyl glucosamine, rich medium, and rich medium supplemented with 5 g/L *N*-acetyl glucosamine.

involved in the tricarboxylic acid (TCA) cycle (*Figure 2B and C*) and the Embden–Meyerhof–Parnas (EMP) glycolysis pathway (*Figure 2—figure supplement 1A, B*) (which both contribute to energy production) were upregulated. In addition, the expression of genes encoding NADH-ubiquinone oxidoreductase and flagellum assembly-related proteins were also upregulated (*Figure 2D and E*). To verify the transcriptomic data, we performed real-time quantitative reverse transcription PCR (qRT-PCR) assays, which showed the same gene expression variation, consistent with the transcriptomic results (*Figure 2—figure supplement 2*). Based on the combined results of the growth assay and transcriptomic analyses, we concluded that strain ZRK32 growth is better when cultured using nutrient-rich medium.

## *P. heterotrophicis* ZRK32 replicates using a budding mode of division

After reviewing more than 600 TEM photos, we confirmed that strain ZRK32 divided by budding — a method also reported in other *Planctomycetes* bacteria (*Figure 1C* and *Figure 1—figure supplement 1*; *Wiegand et al., 2020b*). Remarkably, during the early stages of budding in strain ZRK32, the extracellular membrane extended and formed a bulge, which grew until it was a similar size to the mother cell (*Figure 3A and B*, panels 1–4). The genetic materials within the nucleoid then duplicated and divided equally between the mother and daughter cells, along with other cytoplasmic contents (*Figure 3A and B*, panels 5–8). The daughter cell then separated from the mother cell, completing cell division.

Next, we investigated whether genes associated with budding method of cell division were present in the genome of strain ZRK32, and whether they were functional during bacterial growth. We performed genomic and transcriptomic analyses of strain ZRK32. We did not detect the cell division protein, FtsZ, in strain ZRK32, but we did identify other Fts-related proteins (e.g. FtsJ, FtsK, FtsL, FtsW, FtsX, and FtsY) and rod shape-determining proteins (MreB, MreC, and MreD); the expressions of genes encoding these Fts-related proteins were upregulated in the ZRK32 strains cultured in the rich medium compared with strains cultured in basal medium (*Figure 3C*).

## Effects of $NO_3^-$, $NH_4^+$, and $NO_2^-$ on *P. heterotrophicis* ZRK32 growth

As *Planctomycetes* bacteria are involved in nitrogen cycling, we tested the effects of different nitrogen-containing substances (including $NO_3^-$, $NH_4^+$, and $NO_2^-$) on strain ZRK32 growth. These assays showed that adding $NO_3^-$ or $NH_4^+$ to the culture medium increased strain ZRK32 growth, while adding $NO_2^-$ inhibited growth (*Figure 4A*). The concentration of $NO_3^-$ decreased from ~21 mM to ~6 mM, and then to ~4 mM after strain ZRK32 had been incubating for 4 days and 6 days, respectively. The concentration of $NH_4^+$ increased from ~0 mM to ~11 mM, and then to ~7 mM after strain ZRK32 had been incubating for 4 days and 6 days, respectively. These results strongly suggest that strain ZRK32 can effectively convert $NO_3^-$ to $NH_4^+$ (*Figure 4B*). In addition, when strain ZRK32 was incubated in the rich medium supplemented with $NH_4^+$ for 4 days and 6 days, the concentration of $NH_4^+$ decreased from ~19 mM to ~11 mM, and then to ~4 mM, with no change in the concentrations of $NO_3^-$ and $NO_2^-$ (*Figure 4C*). To investigate nitrogen metabolism in strain ZRK32, we analyzed the strain ZRK32 genome and found that it contained a complete nitrate reduction pathway and key genes responsible for the conversion of ammonia to glutamate (*Figure 4D*), which explains the results in *Figure 4B and C*. Subsequently, we performed transcriptome sequencing analysis and found that the genes encoding nitrate reductase (NapA and NapB), nitrite reductase (NirB), glutamine synthetase, and glutamate synthase were all simultaneously upregulated in the presence of $NO_3^-$. We also found that the genes encoding glutamine synthetase and glutamate synthase were upregulated in the presence of $NH_4^+$ (*Figure 4E*). However, no differential expression was observed for the gene, *nirD,* in the presence of $NO_3^-$ or $NH_4^+$, even though this gene encodes a nitrite reductase. Moreover, we observed that some genes involved in the TCA cycle (*Figure 4F*), the EMP glycolysis pathway (*Figure 2—figure supplement 1C*), and genes encoding NADH-ubiquinone oxidoreductase-related proteins (*Figure 4G*) were upregulated in the presence of $NO_3^-$ or $NH_4^+$, but downregulated in the presence of $NO_2^-$. We also observed that the expressions of many genes associated with flagellum assembly were upregulated when $NO_3^-$ or $NH_4^+$ were supplemented into the culture medium (*Figure 4H*). We observed similar trends in our qRT-PCR results (*Figure 4—figure supplement 1*), which validated our RNA-seq results.

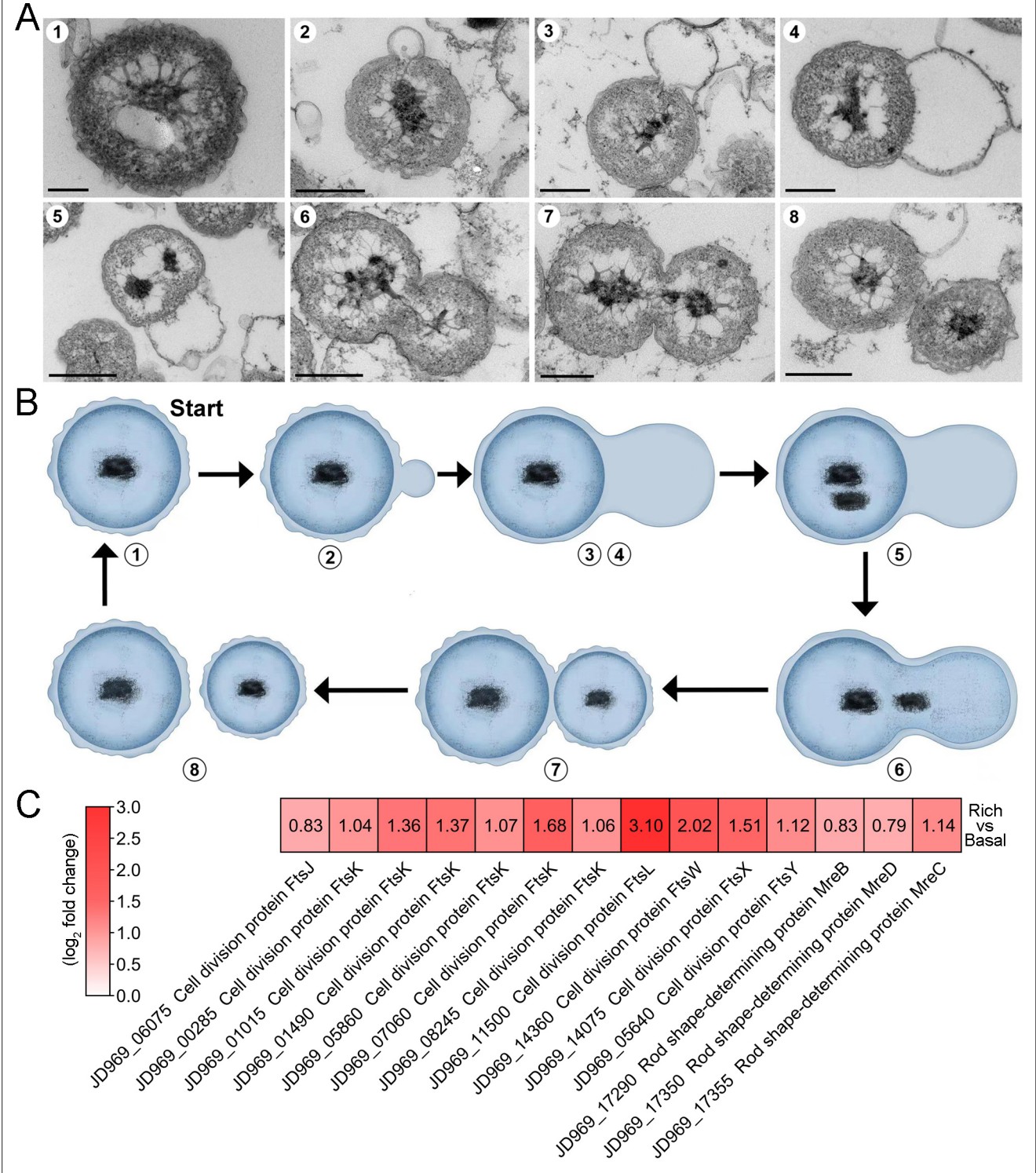

**Figure 3.** The mode of cell division utilized by *P. heterotrophicis* ZRK32. (**A**) Ultrathin transmission electron microscope (TEM) sections showing the process of polar budding division (panels 1–8) in strain ZRK32. Images representing the different phases of cell division are shown. (**B**) The proposed model of cell division of strain ZRK32 based on the TEM observation shown in panel B. The numbers in panels A and B correspond to the same phase of division. (**C**) Transcriptomics-based heat map showing the differentially expressed genes that encode different key proteins associated with cell division in strain ZRK32. The numbers in panel A represent the fold change of gene expression (by using the $\log_2$ value). Scale bars = 200 nm.

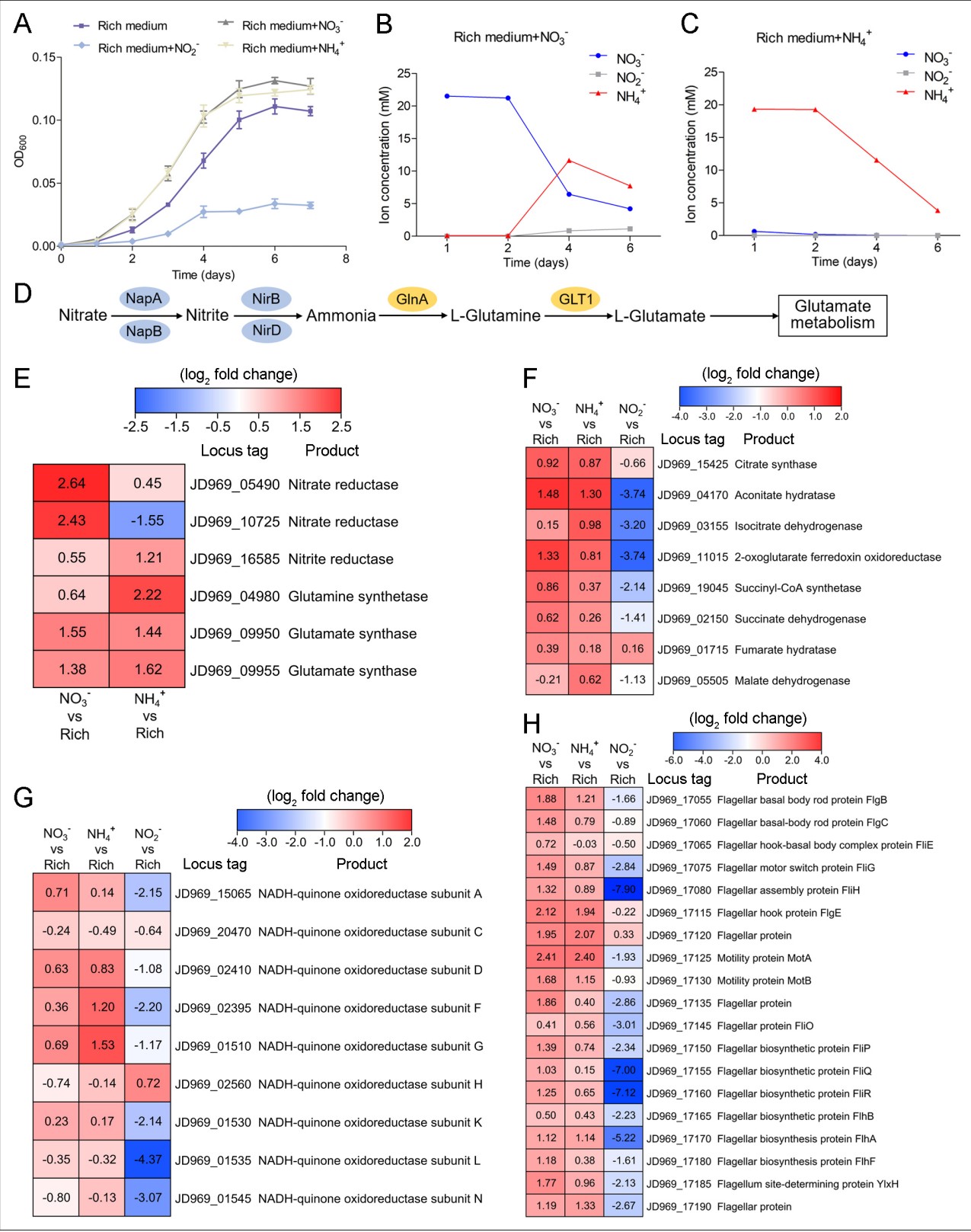

**Figure 4.** Nitrogen metabolism assays of *P. heterotrophicis* ZRK32. (**A**) Growth curves of ZRK32 strains cultivated in the rich medium alone and cultivated in rich medium supplemented with either 20 mM $NO_3^-$, 20 mM $NH_4^+$, or 20 mM $NO_2^-$. Data shown as mean; error bars = SD (Standard Deviation). (**B**) The dynamics of concentrations of $NO_3^-$, $NH_4^+$, and $NO_2^-$ in strains of ZRK32 cultivated in the rich medium supplemented with 20 mM $NO_3^-$. (**C**) The dynamics of concentrations of $NO_3^-$, $NH_4^+$, and $NO_2^-$ in strains of ZRK32 cultivated in the rich medium supplemented with 20 mM $NH_4^+$. (**D**) The predicted nitrogen

*Figure 4 continued on next page*

*Figure 4 continued*

metabolism pathway of strain ZRK32. Abbreviations: NapA, periplasmic nitrate reductase; NapB, periplasmic nitrate reductase, electron transfer subunit; NirB, nitrite reductase (NADH) large subunit; NirD, nitrite reductase (NADH) small subunit; GlnA, glutamine synthetase; GLT1, glutamate synthase. Transcriptomics-based heat map showing the relative expression levels of genes associated with nitrogen metabolism (**E**), the tricarboxylic acid (TCA) cycle (**F**), NADH-quinone oxidoreductase (**G**), and flagellar assembly (**H**) in strains of ZRK32 cultivated in the rich medium supplemented with different inorganic nitrogen sources (20 mM $NO_3^-$, 20 mM $NH_4^+$, or 20 mM $NO_2^-$) compared with strains cultivated in the rich medium alone. 'Rich' indicates rich medium. '$NO_3^-$, $NH_4^+$, and $NO_2^-$' indicate rich medium supplemented with 20 mM $NO_3^-$, 20 mM $NH_4^+$, and 20 mM $NO_2^-$, respectively. The numbers in panels E, F, G, and H represent the fold change of gene expression (by using the $\log_2$ value).

The online version of this article includes the following figure supplement(s) for figure 4:

**Figure supplement 1.** Real-time quantitative reverse transcription PCR (qRT-PCR) detection of the relative expression levels of the genes associated with nitrogen metabolism (**A**), tricarboxylic acid (TCA) cycle (**B**), NADH-quinone oxidoreductase (**C**), flagellar assembly (**D**), and Embden–Meyerhof–Parnas (EMP) glycolysis pathway (**E**) of ZRK32 strains cultivated in the rich medium supplemented with different inorganic nitrogen sources (20 mM $NO_3^-$, 20 mM $NH_4^+$, or 20 mM $NO_2^-$) compared with strains cultivated in the rich medium alone.

**Figure supplement 2.** Muti-omics-based central metabolism model of *P. heterotrophicis* ZRK32.

## $NO_3^-$ and $NH_4^+$ induce the release of a chronic bacteriophage in *P. heterotrophicis* strain ZRK32

Bacteriophages are widely distributed across oceans and can regulate nitrogen metabolism in their host (*Cassman et al., 2012*; *Monier et al., 2017*; *Gazitúa et al., 2021*; *Wang et al., 2022*). We therefore investigated whether bacteriophages affected nitrogen metabolism in strain ZRK32. TEM observations showed that phage-like structures (hexagonal phages, ~30 nm) were present in cell suspensions of the ZRK32 strain that had been cultured using nutrient-rich medium supplemented with either $NO_3^-$ or $NH_4^+$ (*Figure 5A*, panels II and III). In contrast, no phage-like structures were observed in the cell suspensions of ZRK32 strain that was cultured in the rich medium alone (without $NO_3^-$ or $NH_4^+$) (*Figure 5A*, panel I). This suggests that the presence of $NO_3^-$ or $NH_4^+$ stimulated the release of the bacteriophages from strain ZRK32. Most chronic bacteriophages do not negatively affect their host's growth when cultivated in the laboratory (*Alarcón-Schumacher et al., 2022*). Consistently, the replication and release of the bacteriophages from strain ZRK32 did not kill the host cell, consistent well with the key feature of chronic bacteriophages (*Howard-Varona et al., 2017*). By comparing genomic sequences, we confirmed that the genome of the phage induced by $NO_3^-$ was the same as the phage induced by $NH_4^+$. When we compared this phage genome (Phage-ZRK32, 21.9 kb) (*Figure 5B*) with the host genome (strain ZRK32) (using Galaxy Version 2.6.0 [https://galaxy.pasteur.fr/] [*Afgan et al., 2018*] with the NCBI BLASTN method), we found that the Phage-ZRK32 genome was outside of the host chromosome; this indicates that this chronic bacteriophage is extrachromosomal, which is consistent with previous reports (*Chevallereau et al., 2022*). In addition to the genes encoding phage-associated proteins, the genome of Phage-ZRK32 also has many AMGs, which encode glutamine amidotransferase, amidoligase, glutathione synthase, and gamma-glutamylcyclotransferase (*Figure 5B*). Although, some genes (including genes encoding amidoligase, glutathione synthase, and gamma-glutamylcyclotransferase) were absent from the strain ZRK32 genome.

To verify whether these AMGs were functional, we investigated whether Phage-ZRK32 was capable of reprogramming nitrogen metabolism and promoting growth in other marine bacteria. We selected the aerobic marine bacterium, *P. stutzeri* 273 (*Wu et al., 2016*), and examined the effects of Phage-ZRK32 on its growth. This showed that Phage-ZRK32 promoted *P. stutzeri* 273 growth by facilitating the metabolism and utilization of $NO_3^-$ and $NH_4^+$ (*Figure 5C and D*). In particular, adding Phage-ZRK32 and $NH_4^+$ into the oligotrophic medium resulted in an approximate three- to eightfold increase in growth compared with strains cultivated without $NH_4^+$ or Phage-ZRK32 supplementation (*Figure 5D*). The AMGs that encode amidoligase and gamma-glutamylcyclotransferase were absent from the *P. stutzeri* 273 genome, even though it contains a complete nitrate reduction pathway and some genes responsible for converting ammonia to glutamate. Therefore, the Phage-ZRK32 AMGs might facilitate nitrogen metabolism and amino acid generation in *P. stutzeri* 273 in a similar way to *P. heterotrophicis* ZRK32.

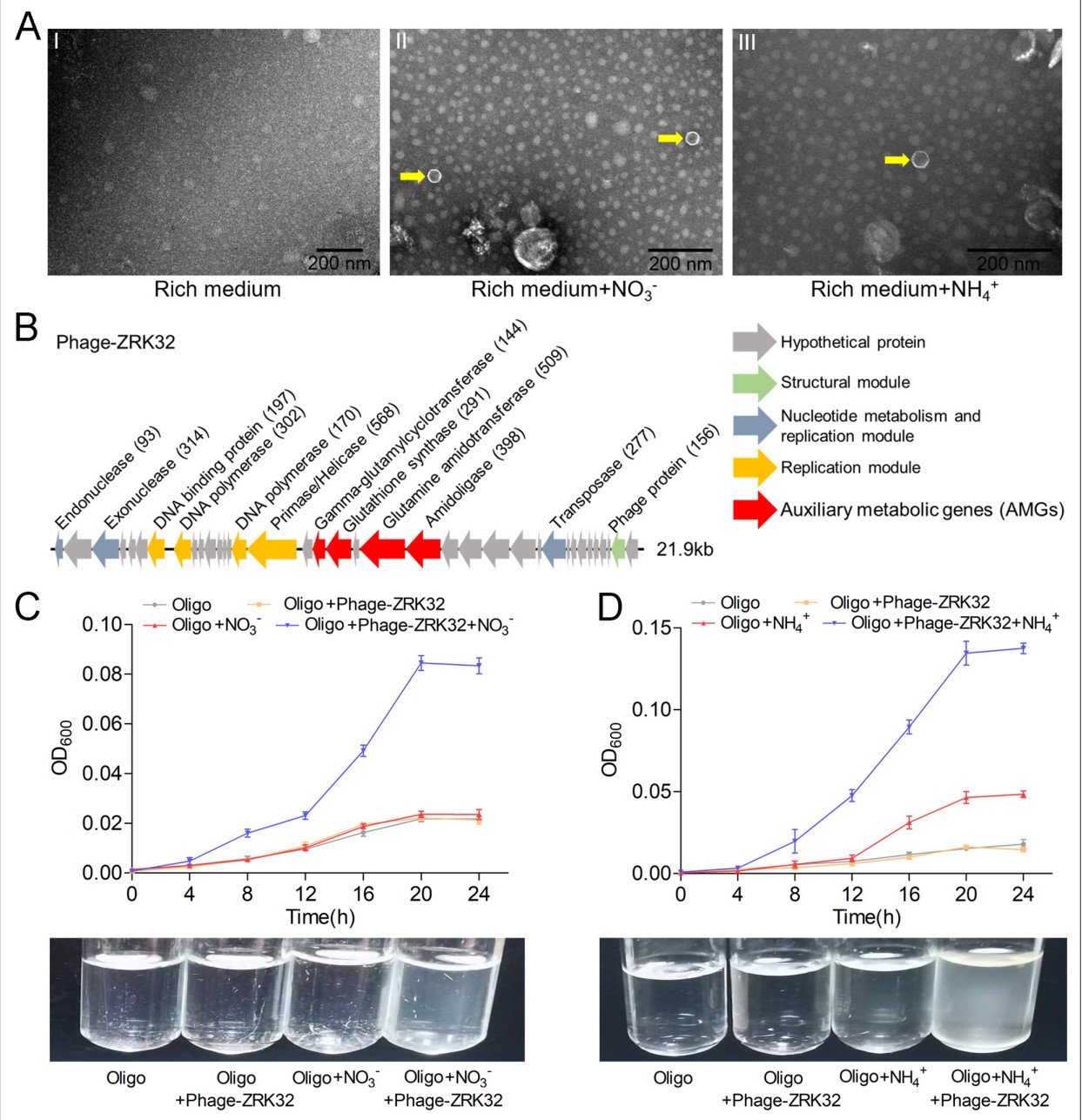

**Figure 5.** Observation and functional assay of the chronic bacteriophage induced by NO$_3^-$ or NH$_4^+$ from *P. heterotrophicis* ZRK32. (**A**) Transmission electron microscope (TEM) observation of phages extracted from the cell suspensions of ZRK32 strains that cultured in either the rich medium alone, or rich medium supplemented with 20 mM of either NO$_3^-$ or NH$_4^+$. (A, panel I) No phage-like particles were observed in the cell suspensions from the ZRK32 strain cultured in the rich medium. (A, panels II and III) Hexagonal phages (indicated with yellow arrows) observed in the cell suspensions from the ZRK32 strains cultured in the rich medium supplemented with 20 mM of either NO$_3^-$ or NH$_4^+$. Scale bars: 200 nm. (**B**) A diagram showing the genomic composition of Phage-ZRK32. The arrows represent different ORFs and the direction of transcription. The main putative gene products of this phage are shown, and the numbers in brackets indicate the numbers of amino acids. Hypothetical proteins are indicated by gray arrows, structural modules are indicated by green arrows, nucleotide metabolism is indicated by blue-gray arrows, the replication module is indicated by gold arrows, and auxiliary metabolic genes (AMGs) are indicated by red arrows. The size of the phage genome is shown beside the gene cluster. (**C**) Bacterial growth curve showing the growth rate of strains of *Pseudomonas stutzeri* 273 cultivated in either oligotrophic medium, oligotrophic medium supplemented with Phage-ZRK32, oligotrophic medium supplemented with 20 mM NO$_3^-$, or oligotrophic medium supplemented with 20 mM NO$_3^-$ and Phage-ZRK32. (**D**) Bacterial growth curve showing the growth rate of strains of *P. stutzeri* 273 cultivated in either oligotrophic medium, oligotrophic medium supplemented with Phage-ZRK32, oligotrophic medium supplemented with 20 mM NH$_4^+$, or oligotrophic medium supplemented with 20 mM NH$_4^+$ and Phage-ZRK32. 'Oligo' indicates oligotrophic medium. Data shown as mean; error bars = SD (Standard Deviation).

*Figure 5 continued on next page*

*Figure 5 continued*

The online version of this article includes the following figure supplement(s) for figure 5:

**Figure supplement 1.** Phylogenetic analysis of Phage-ZRK32, some related phages, and bacterial hosts, based on the aligned amino acid sequences of amidoligase.

**Figure supplement 2.** Phylogenetic analysis of Phage-ZRK32, some related phages, and bacterial hosts, based on the aligned amino acid sequences of glutamine amidotransferase.

**Figure supplement 3.** Phylogenetic analysis of Phage-ZRK32, some related phages, and bacterial hosts, based on the aligned amino acid sequences of gamma-glutamylcyclotransferase.

**Figure supplement 4.** Phylogenetic analysis of Phage-ZRK32, some related phages, and bacterial hosts, based on the aligned amino acid sequences of glutathione synthase.

## Discussion

Until recently, most research on *Planctomycetes* has focused on strains in freshwater and shallow ocean environments (*Bondoso et al., 2011*; *Kallscheuer et al., 2020*), with few studies on deep-sea strains; this is likely due to the logistical difficulties associated with sampling and cultivating these strains. The availability of cultured and characterized representatives for many phylogenetic clades within the phylum Planctomycetes are lacking (*Dedysh et al., 2021*). Therefore, more approaches and media types, like those used in our study (*Figure 1A*), should be developed to obtain *Planctomycetes* bacteria from different deep-sea environments. The vast majority of *Planctomycetes* members have been isolated using an oligotrophic medium supplemented with *N*-acetyl glucosamine (*Kaboré et al., 2020*; *Peeters et al., 2020*; *Salbreiter et al., 2020*; *Wiegand et al., 2020a*); use of this medium previously resulted in the breakthrough isolation of 79 *Planctomycetes* strains (*Wiegand et al., 2020b*). However, we observed that strains of *P. heterotrophicis* ZRK32 grew much better when cultivated in a rich medium compared with strains cultivated in the basal medium (*Figure 2A*). We also found that *N*-acetyl glucosamine did not stimulate the growth of strain ZRK32 (*Figure 2—figure supplement 3*). These findings have revealed that strain ZRK32 prefers a nutrient-rich medium, which is different from most other *Planctomycetes* bacteria (*Wiegand et al., 2018*; *Wiegand et al., 2020b*).

Notably, when growing in the rich medium, the expressions of most genes involved in the TCA cycle and EMP glycolysis pathway in strain ZRK32 were upregulated (*Figure 2B–D*, *Figure 2—figure supplement 1B*, and *Figure 2—figure supplement 2*), suggesting that strain ZRK32 might function through the complete TCA metabolic pathway and EMP glycolysis pathway to obtain energy for growth (*Figure 4—figure supplement 2*; *Zheng et al., 2021a*). Consistent with the presence of EMP glycolysis pathway in strain ZRK32, we found that it could use a variety of sugars including glucose, maltose, fructose, isomaltose, galactose, D-mannose, and rhamnose (*Supplementary file 2*). As for the presence of TCA cycle in the anaerobic strain ZRK32, we propose that other alternative electron acceptors (such as sulfate reducers, nitrate reducers, iron reducers, etc.) may be used instead of oxygen molecule as the final electron acceptor in the electron transfer chain, as shown in other anaerobic bacteria (*Alteri et al., 2012*).

Strain ZRK32 possesses a condensed and intact nucleoid-like structure and a complex membrane structure (*Figure 1D*), which is similar to other reported *Planctomycetes* bacteria (*Kaushik et al., 2020*; *Dedysh et al., 2021*). In addition, TEM observation of ultrathin sections of cells from strain ZRK32 showed some eukaryote-like structures (*Figure 1—figure supplement 5*, panels 1–6). Although it is impossible to judge accurately what these structures are based on current methods, the internal structures of *Planctomycetes* bacteria are nevertheless fascinating. We also observed some vesicle-like (*Figure 1—figure supplement 5*, panels 7–8) and vacuole-like structures (*Figure 1—figure supplement 5*, panel 9) in strain ZRK32, similar to those observed in other *Planctomycetes* members (*Wiegand et al., 2018*). Vacuoles store cellular components (such as proteins and sugars, etc.) and play essential roles in plant responses to different biotic/abiotic signaling pathways (*Zhang et al., 2014*). The presence of vacuoles in strain ZRK32 suggests that *Planctomycetes* bacteria might have adopted a eukaryotic mechanism for nutrient metabolism and signal transduction.

Most species within the class Planctomycetia divide by budding, and species within the class Phycisphaerae divide by binary fission (*Fukunaga et al., 2009*; *Yoon et al., 2014*; *Kovaleva et al., 2015*; *Peeters et al., 2020*; *Pradel et al., 2020*; *Salbreiter et al., 2020*; *Wiegand et al., 2020b*). However, the *Poriferisphaera* strain ZRK32 (*Figure 1C* and *Figure 1—figure supplement 1*) was demonstrated

to divide by budding, similar to the *Poriferisphaera* strain KS4 proposed by *Kallscheuer et al., 2020*, suggesting that members within the class Phycisphaerae might divide through both binary fission and budding. It is noteworthy that *P. heterotrophicis* ZRK32 forms an empty cell framework first, followed by the entry of the cellular contents (*Figure 3A and B*), which is different from the typical budding mode observed in yeast, where the cellular framework and contents both extend simultaneously (*Herskowitz, 1988*). During microbial cell division, members of the Fts family of proteins (tubulin homologs) usually assemble at the future site of cell division, forming a contractile ring known as the Z ring (*Wiegand et al., 2018*). We observed that the expressions of numerous Fts-related proteins (FtsJ, FtsK, FtsL, FtsW, FtsX, and FtsY) in strain ZRK32 were upregulated when the strain had been cultivated in the rich medium (*Figure 3C*). However, the FtsZ protein was absent from strain ZRK32, which is consistent with the proposal that the *ftsZ* gene is absent from *Planctomycetes* genomes (*Jogler et al., 2012*). In addition, we also found that some genes that encoded rod shape-determining proteins (MreB, MreC, and MreD) were present in strain ZRK32, and that their expressions were upregulated in the strains cultivated in the rich medium (*Figure 3C*). MreB, usually participates in the formation and degradation of peptidoglycan, ultimately determines bacterial cell shape (*Rohs and Bernhardt, 2021*).

*Planctomycetes* bacteria are major players in the global nitrogen cycling and perform important reactions, such as the anaerobic ammonium oxidation process that oxidizes $NH_4^+$ to $N_2$ gas, using $NO_2^-$ as an electron acceptor (*Strous et al., 1998*; *Oshiki et al., 2016*; *Wiegand et al., 2018*). To date, most studies on anaerobic ammonium oxidation have been related to the monophyletic group (*Candidatus* Brocadiae) in the phylum Planctomycetes (*Strous et al., 1999*), with only few reports on the process of nitrogen metabolism in other *Planctomycetes* bacteria. Here, we found that $NO_3^-$ and $NH_4^+$ promoted the growth of strain ZRK32, while $NO_2^-$ inhibited its growth (*Figure 4A*). Based on our data and results from others, we speculate that $NO_3^-$ might act as a terminal electron acceptor in the respiratory electron transport chain of strain ZRK32. In addition, $NO_2^-$ might react with the iron-sulfur proteins in strain ZRK32 to form iron-nitric oxide complexes, which then inactivate iron-sulfur enzymes and inhibit the growth of strain ZRK32 (*Reddy et al., 1983*). We speculate that strain ZRK32 converts $NO_3^-$ to $NH_4^+$ (*Figure 4B and C*), enabling the $NH_4^+$ created to enter the glutamate metabolic pathway (*Figure 4D and E*) — a pathway that is closely associated with several processes, including nitrogen metabolism, the TCA cycle, the EMP glycolysis pathway, and amino acid metabolism (*Figure 4—figure supplement 2*). Consistently, in the presence of $NO_3^-$ or $NH_4^+$, genes associated with the TCA cycle and the EMP glycolysis pathway were upregulated (*Figure 4F*, *Figure 2—figure supplement 1C*, and *Figure 4—figure supplement 1*). The presence of rich nutrients, and either $NO_3^-$ or $NH_4^+$ also stimulated the expression of genes encoding the NADH-ubiquinone oxidoreductase complex (*Figures 2D and 4G*). This complex couples the oxidation of NADH and the reduction of ubiquinone to generate a proton gradient, which is then used for ATP synthesis (*Reda et al., 2008*). Notably, large number of genes associated with flagellum assembly were also upregulated in strain ZRK32 (*Figures 2E and 4H*). Flagellum-mediated motility is beneficial for bacteria, not only because it allows them to respond quickly to an ever-changing environment but also because it enables them to seek and acquire nutrients for survival (*Wadhams and Armitage, 2004*; *Zheng et al., 2021b*). Thus, strain ZRK32 might regulate the formation and motility of its flagellum to accelerate the absorption and utilization of nutrients available in its environment.

One of our most exciting results was that $NO_3^-$ or $NH_4^+$ could induce the release of a chronic bacteriophage (Phage-ZRK32) from strain ZRK32 (*Figure 5A*). The Phage-ZRK32 was capable of facilitating nitrogen metabolism and amino acid metabolism in strain ZRK32 through the function of AMGs, which incorporate nitrogen into certain amino acids (including glutamate, cysteine, and glycine) (*Figure 5B*; *Orlowski and Meister, 1970*; *Mouilleron and Golinelli-Pimpaneau, 2007*; *Iyer et al., 2009*; *Lu, 2013*). Phylogenetic analyses were performed (*Figure 5—figure supplements 1–4*) using AMG amino acid sequences (including amidoligase, glutamine amidotransferase, gamma-glutamylcyclotransferase, and glutathione synthase) from Phage-ZRK32, in addition to the same amino acid sequences from related phages and their bacterial hosts. The results showed that these AMGs may have been acquired from *Pseudomonas* via horizontal gene transfer (*Matilla et al., 2014*; *Zhou et al., 2019*). Consistently, Phage-ZRK32 promoted the growth of a marine *Pseudomonas* bacterium (*P. stutzeri* 273) in the presence of either $NO_3^-$ or $NH_4^+$ (*Figure 5C and D*). Given that the AMGs encoding amidoligase and gamma-glutamylcyclotransferase were absent in the genome of *P. stutzeri* 273, we speculate that

Phage-ZRK32 might promote the growth of *P. stutzeri* 273 by facilitating nitrogen metabolism and amino acid generation, as in the *Planctomycetes* strain ZRK32. Most bacteriophage life cycles are described as a lytic or lysogenic cycle (*Du Toit, 2017*). Currently, more attention has been given to the chronic life cycle, where bacterial growth continues despite phage reproduction (*Hoffmann Berling and Maze, 1964*), and the progeny of these phage particles are released from host cells via extrusion or budding without killing the host (*Putzrath and Maniloff, 1977*; *Russel, 1991*; *Marvin et al., 2014*). Undoubtedly, Phage-ZRK32, which was induced by either $NO_3^-$ or $NH_4^+$ in *Planctomycetes* strain ZRK32, is a chronic bacteriophage. Moreover, it has recently been reported that the tailless Caudoviricetes phage particles are enclosed in lipid membrane and are released from the host cells by a nonlytic mechanism (*Liu et al., 2022*), and the prophage induction contributes to the production of membrane vesicles by *Lacticaseibacillus casei* BL23 during cell growth (*da Silva Barreira et al., 2022*). Considering that strain ZRK32 has a large number of membrane vesicles during cell growth (*Figure 1—figure supplement 5*), we speculated that Phage-ZRK32 might be a membrane vesicle-engulfed phage and its release should be related to membrane vesicles. Altogether, our findings provide a novel insight into nitrogen metabolism in *Planctomycetes* bacteria and provide a suitable model to study the interactions between *Planctomycetes* and viruses.

# Materials and methods

## Enrichment and cultivation of deep-sea *Planctomycetes* bacteria

To isolate and cultivate *Planctomycetes* bacteria, 2 g deep-sea sediment samples collected by *RV KEXUE* from a typical cold seep in the South China Sea (E119°17'07.322", N22°06'58.598") at a depth of ~1143 m were added to a 500 mL anaerobic bottle containing 400 mL basal medium (1.0 g/L yeast extract, 20.0 g/L NaCl, 1.0 g/L CH$_3$COONa, 1.0 g/L NaHCO$_3$, 0.5 g/L KH$_2$PO$_4$, 0.2 g/L MgSO$_4$.7H$_2$O, 0.7 g/L cysteine hydrochloride, 500 µL/L 0.1% (wt/vol) resazurin, 1.0 L sterilized distilled water, pH 7.0) supplemented with 1.0 g/L NH$_4$Cl and 1.0 g/L NaNO$_3$, ensuring headspace volume was retained above the liquid surface. The medium was prepared under a 100% N$_2$ gas phase and sterilized by autoclaving at 115°C for 30 min; following this, rifampicin (100 µg/mL) was added. The inoculated media were anaerobically incubated at either 4°C or 28°C for 1 month. The basal medium supplemented with 1.0 g/L NH$_4$Cl, 1.0 g/L NaNO$_3$, and 15 g/L agar was evenly spread on to the inside wall of a Hungate tube, which formed a thin layer of medium for the bacteria to grow. After this, 50 µL of the enriched culture was anaerobically transferred into an anaerobic roll tube and then spread on to the medium layer. These tubes were also anaerobically cultured at either 4°C or 28°C for 10 days. Single colonies growing at 28°C were selected using sterilized bamboo sticks; they were then cultured in the 15 mL Hungate tube containing 10 mL basal medium (supplemented with 1.0 g/L NH$_4$Cl and 1.0 g/L NaNO$_3$) at 28°C for 7 days under a 100% N$_2$ atmosphere. One strain was identified as a member of the phylum Planctomycetes, but was noted to have less than 98% 16S rRNA gene sequence similarity to other cultured strains; this strain was therefore selected to be cultivated. Strain ZRK32 was selected and purified by repeating the Hungate roll-tube method. The purity of strain ZRK32 was confirmed regularly by observation using a TEM and by repeating partial sequencing of the 16S rRNA gene. As strain ZRK32 grew slowly in basal medium, we used a rich culture medium (10.0 g/L yeast extract, 20.0 g/L NaCl, 1.0 g/L CH$_3$COONa, 1.0 g/L NaHCO$_3$, 0.5 g/L KH$_2$PO$_4$, 0.2 g/L MgSO$_4$.7H$_2$O, 0.7 g/L cysteine hydrochloride, 500 µL/L 0.1% [wt/vol] resazurin, 1.0 L sterilized distilled water, pH 7.0).

## TEM observation

To observe the morphological characteristics of strain ZRK32, 10 mL culture was collected by centrifuging at 5000×*g* for 10 min. Cells were then washed three times with PBS buffer (137 mM NaCl, 2.7 mM KCl, 10 mM Na$_2$HPO$_4$, 1.8 mM KH$_2$PO$_4$, 1 L sterile water, pH 7.4). Finally, the cells were suspended in 20 µL PBS buffer, and then transferred onto copper grids coated with a carbon film by immersing the grids in the cell suspension for 30 min (*Zheng et al., 2021a*). To observe the ultrastructure of strain ZRK32, ultrathin sections were prepared using methods previously described (*Graham and Orenstein, 2007*). Briefly, 500 mL of cells (cultured for 6 days at 28°C) were collected by centrifuging at 5000×*g* for 20 min, and then washing three times with PBS buffer. The cells were then preserved in 2.5% (vol/vol) glutaraldehyde for 12 hr at 4°C and then dehydrated using different ethanol concentrations (30%, 50%, 70%, 90%, and 100%) for 10 min each time. The cells were then

embedded in a plastic resin. Finally, 50–70 nm ultrathin sections were produced using an ultramicro-tome (Leica EM UC7, Germany) and then stained using uranyl acetate and lead citrate. All samples were examined under TEM (HT7700, Hitachi, Japan).

## Genome sequencing, annotation, and analysis of strain ZRK32

For genomic sequencing, strain ZRK32 was grown in the liquid-rich medium and harvested after 1 week of incubation at 28°C. Genomic DNA was isolated by using the PowerSoil DNA isolation kit (Mo Bio Laboratories Inc, Carlsbad, CA, USA). Thereafter, the genome sequencing was carried out with both the Illumina NovaSeq PE150 (San Diego, CA, USA) and Nanopore PromethION platform (Oxford, UK) at the Beijing Novogene Bioinformatics Technology Co., Ltd. A complete description of the library construction, sequencing, and assembly was performed as previously described (*Zheng et al., 2021a*). We used seven databases to predict gene functions, including Pfam (Protein Families Database, http://pfam.xfam.org/), GO (Gene Ontology, http://geneontology.org/) (*Ashburner et al., 2000*), KEGG (Kyoto Encyclopedia of Genes and Genomes, http://www.genome.jp/kegg/) (*Kanehisa et al., 2004*), COG (Clusters of Orthologous Groups, http://www.ncbi.nlm.nih.gov/COG/) (*Galperin et al., 2015*), NR (Non-Redundant Protein Database databases), TCDB (Transporter Classification Database), and Swiss-Prot (http://www.ebi.ac.uk/uniprot/) (*Bairoch and Apweiler, 2000*). A whole genome Blast search (E-value less than 1e-5, minimal alignment length percentage larger than 40%) was performed against above seven databases.

In addition, the genome relatedness values were calculated by multiple approaches, including ANI based on the MUMMER ultra-rapid aligning tool (ANIm) and the BLASTN algorithm (ANIb), the Tetra, and *is*DDH similarity. ANIm, ANIb, and Tetra values were calculated using the JSpecies WS (http://jspecies.ribohost.com/jspeciesws/) (*Richter et al., 2016*). The recommended species criterion cut-offs were used: 95% for the ANIb and ANIm, 0.99 for the Tetra signature. The *is*DDH similarity values were calculated by the Genome-to-Genome Distance Calculator (GGDC) (http://ggdc.dsmz.de/) (*Meier-Kolthoff et al., 2013*). A value of 70% *is*DDH similarity was used as a recommended standard for delineating species.

## Phylogenetic analysis

To construct a maximum likelihood 16S rRNA phylogenetic tree, the full-length 16S rRNA gene sequences of strain ZRK32 and other related taxa were extracted from their corresponding genomes (https://www.ncbi.nlm.nih.gov/). The maximum likelihood genome phylogenetic tree was constructed from a concatenated alignment of 37 protein-coding genes (*Wu et al., 2013*) (extracted from the genomes using Phylosift v1.0.1; *Darling et al., 2014*); all genes were present in a single copy and were universally distributed in both archaea and bacteria (*Supplementary file 3*). The reference sequences of four AMGs encoding amidoligase, glutamine amidotransferase, gamma-glutamylcyclotransferase, and glutathione synthase were retrieved by blasting the phage gene against the entire NCBI database, respectively. All phylogenetic trees were constructed using the W-IQ-TREE web server (http://iqtree.cibiv.univie.ac.at) (*Trifinopoulos et al., 2016*) using the 'GTR+F+I+G4' model, and the Interactive Tree of Life (iTOL v5) online tool (*Letunic and Bork, 2021*) was used to edit the phylogenetic trees.

## Physiological tests

Effects of temperature, pH, and NaCl concentration on the growth of strain ZRK32 were determined in the rich medium as described above. To evaluate the temperature range for growth, cultures were incubated at 4°C, 16°C, 24°C, 28°C, 32°C, 37°C, 45°C, 60°C (pH 7.0). To determine the pH range for growth, the medium was adjusted at optimum temperature (28°C) to pH 4.0–10.0 with increments of 0.5 pH units under a 100% $N_2$ atmosphere. NaCl requirements were tested in the modified rich medium (without 20.0 g/L NaCl) supplemented with 0–10% (wt/vol) NaCl (1.0% intervals). Single sugar (including glucose, maltose, fructose, sucrose, starch, isomaltose, trehalose, galactose, cellulose, xylose, D-mannose, and rhamnose) was added from sterile filtered stock solutions to the final concentration at 20 mM, respectively. Cell culture containing only 0.02 g yeast extract ($L^{-1}$) without adding any other substrates was used as a control. These cultures were incubated at 28°C for 14 days and then the $OD_{600}$ values were measured via a microplate reader (Infinite M1000 Pro; Tecan, Mannedorf, Switzerland). For each experiment, three biological replicates were performed.

## Growth assays of strain ZRK32

To assess the effects of nutrient-rich media on strain ZRK32 growth, we set up different cultures using either rich medium or basal medium. To assess the effects of different inorganic nitrogen sources (20 mM $NO_3^-$, 20 mM $NH_4^+$, and 20 mM $NO_2^-$) on strain ZRK32 growth, we used a rich culture medium supplemented with the nitrogen sources mentioned above. For each growth assay, 15 mL of strain ZRK32 culture was inoculated in a 2 L Hungate bottle containing 1.5 L of the respective media. All Hungate bottles were anaerobically incubated at 28°C. Bacterial growth was monitored by measuring daily $OD_{600}$ values via a microplate reader until cell growth reached a stationary phase. Three replicates were performed for each condition. The concentrations of $NO_3^-$, $NH_4^+$, and $NO_2^-$ were determined using a continuous flow analyzer (SKALAR-SAN++, Netherlands), which has an analytical precision of 6%.

## Transcriptomics

For transcriptomic sequencing, strains of ZRK32 were cultured in 1.5 L of either basal medium, rich medium, or rich medium supplemented with different nitrogen sources (20 mM $NO_3^-$, 20 mM $NH_4^+$, or 20 mM $NO_2^-$) at 28°C for 6 days. Three biological replicates were cultured for each condition. Following this, cells from the three replicates were mixed and then collected for transcriptomic sequencing by Novogene (Tianjin, China), as previously described (*Zheng et al., 2021a*; *Zheng et al., 2022*). The detailed sequencing information was shown as below: (1) Library preparation for strand-specific transcriptome sequencing. A total amount of 3 µg RNA per sample was used as input material for the RNA sample preparation. Sequencing libraries were generated using NEBNext Ultra Directional RNA Library Prep Kit for Illumina (NEB, USA) following the manufacturer's recommendations and index codes were added to attribute sequences to each sample. Then, rRNA was removed using a specialized kit that left the mRNA. Fragmentation was carried out using divalent cations under elevated temperature in NEBNext First Strand Synthesis Reaction Buffer (5×). First strand cDNA was synthesized using random hexamer primer and M-MuLV Reverse Transcriptase (RNaseH⁻). Second strand cDNA synthesis was subsequently performed using DNA Polymerase I and RNase H. In the reaction buffer, dNTPs with dTTP were replaced by dUTP. Remaining overhangs were converted into blunt ends via exonuclease/polymerase activities. After adenylation of 3′ ends of DNA fragments, NEBNext Adaptor with hairpin loop structure was ligated to prepare for hybridization. In order to select cDNA fragments of preferentially 150–200 bp in length, the library fragments were purified with AMPure XP system (Beckman Coulter, USA). Then, 3 µL USER Enzyme (NEB, USA) was used with size-selected, adaptor-ligated cDNA at 37°C for 15 min followed by 5 min at 95°C before PCR. PCR was performed with Phusion High-Fidelity DNA polymerase, Universal PCR primers, and Index (X) Primer. At last, products were purified (AMPure XP system) and library quality was assessed on the Agilent Bioanalyzer 2100 system. (2) Clustering and sequencing. The clustering of the index-coded samples was performed on a cBot Cluster Generation System using TruSeq PE Cluster Kit v3-cBot-HS (Illumia) according to the manufacturer's instructions. After cluster generation, the library preparations were sequenced on an Illumina HiSeq platform and paired-end reads were generated. (3) Data analysis. Raw data of fastq format were first processed through in-house perl scripts. In this step, clean data were obtained by removing reads containing adapter, reads containing ploy-N, and low quality reads from raw data. At the same time, Q20, Q30, and GC content the clean data were calculated. All the downstream analyses were based on the clean data with high quality. Reference genome and gene model annotation files were downloaded from genome website directly. Both building index of reference genome and aligning clean reads to reference genome were used, Bowtie2-2.2.3 (setting: -D 15 -R 2 -N 0 -L 22 -i S,1,1.15) (*Langmead and Salzberg, 2012*). HTSeq v0.6.1 (default parameters) was used to count the reads numbers mapped to each gene. FPKM of each gene was calculated based on the length of the gene and reads count mapped to this gene. FPKM, expected number of fragments per kilobase of transcript sequence per millions base pairs sequenced, considers the effect of sequencing depth and gene length for the reads count at the same time, and is currently the most commonly used method for estimating gene expression levels (*Trapnell et al., 2009*). (4) Differential expression analysis. Differential expression analysis was performed using the DESeq R package (1.18.0) and edgeR v3.24.3 ($|\log_2$ (fold change)$| \geq 1$ and padj $\leq 0.05$) (*Anders and Huber, 2010*). DESeq provide statistical routines for determining differential expression in digital gene expression data using a model based on the negative binomial distribution. The resulting p-values were adjusted using the Benjamini

and Hochberg's approach for controlling the false discovery rate. Genes with an adjusted p-value <0.05 found by DESeq were assigned as differentially expressed (for DEGSeq without biological replicates). Prior to differential gene expression analysis, for each sequenced library, the read counts were adjusted by edgeR program package through one scaling normalized factor. Differential expression analysis of two conditions was performed using the DEGSeq R package (1.20.0) (*Wang et al., 2010*). The p values were adjusted using the Benjamini and Hochberg method. Corrected p-value of 0.005 and log$_2$ (fold change) of 1 were set as the threshold for significantly differential expression. (5) GO and KEGG enrichment analysis of differentially expressed genes. GO enrichment analysis of differentially expressed genes was implemented by the GOseq R package, in which gene length bias was corrected (*Young et al., 2010*). GO terms with corrected p-value less than 0.05 were considered significantly enriched by differential expressed genes. KEGG is a database resource for understanding high-level functions and utilities of the biological system, such as the cell, the organism, and the ecosystem, from molecular-level information, especially large-scale molecular datasets generated by genome sequencing and other high-throughput experimental technologies (http://www.genome.jp/kegg/) (*Kanehisa et al., 2008*). We used KOBAS software to test the statistical enrichment of differential expression genes in KEGG pathways. All heat maps were made by HemI 1.0.3.3.

## Real-time quantitative reverse transcription PCR

To validate the RNA-seq data, we determined the expression levels of some genes by qRT-PCR. For qRT-PCR, cells of strain ZRK32 cultured in 1.5 L of either basal medium, rich medium, or rich medium supplemented with different nitrogen sources (20 mM $NO_3^-$, 20 mM $NH_4^+$, or 20 mM $NO_2^-$, respectively) at 28°C for 6 days were collected at 8000×$g$ for 20 min. Three biological replicates were cultured for each condition. Total RNA from each sample was extracted using the Trizol reagent (Solarbio, China). The RNA concentration was measured using Spectrophotometer (NanoPhotometer NP80, Implen, Germany). Then, RNAs from corresponding samples were reverse-transcribed into cDNA (complementary DNA) using ReverTra Ace qPCR RT Master Mix with gDNA Remover (TOYOBO, Japan). The transcriptional levels of different genes were determined by qRT-PCR using SYBR Green Realtime PCR Master Mix (TOYOBO, Japan) and the QuantStudio 6 Flex (Thermo Fisher Scientific, USA). The PCR condition was set as follows: initial denaturation at 95°C for 3 min, followed by 40 cycles of denaturation at 95°C for 10 s, annealing at 56°C for 20 s, and extension at 72°C for 20 s. The 16S rRNA gene of strain ZRK32 was used as an internal reference and the gene expression was calculated using the $2^{-\Delta\Delta Ct}$ method (*Livak and Schmittgen, 2001*), with each transcript signal normalized to that of 16S rRNA gene. Transcript signals for each treatment were compared to those of control group. Specific primers for genes associated with the TCA cycle, NADH-ubiquinone oxidoreductase, flagellum assembly, and EMP glycolysis of strain ZRK32 and 16S rRNA gene were designed using Primer 5.0 as shown in *Supplementary file 4*. All qRT-PCR runs were conducted with three biological and three technical replicates.

## Isolation of bacteriophages

Isolation of the bacteriophages was performed using similar methods to those described previously, but with some modifications (*Yamamoto et al., 1970*; *Tseng et al., 1990*; *Kim and Blaschek, 1991*). Strain ZRK32 was inoculated in either rich medium, or rich medium supplemented with 20 mM $NO_3^-$ or 20 mM $NH_4^+$, and then incubated at 28°C for 6 days. Different cultures were collected by centrifuging at 8000×$g$ at 4°C for 20 min; this was repeated three times. The supernatant was filtered through a 0.22 μm Millipore filter (Pall Supor, USA), and then 1 M NaCl was added to lyse the residual bacteria. The supernatant was collected by centrifuging at 8000×$g$ at 4°C for 20 min. The phage particles were immediately precipitated with 100 g/L polyethylene glycol (PEG8000) at 4°C for 2 hr, and collected by centrifuging at 10,000×$g$ at 4°C for 20 min. The phage particles were then suspended in SM buffer (0.01% gelatin, 50 mM Tris-HCl, 100 mM NaCl, and 10 mM $MgSO_4$). The suspension was then extracted three times using an equal volume of chloroform (*Lin et al., 2012*) and collected by centrifuging at 4000×$g$ at 4°C for 20 min. Finally, the clean phage particles were obtained.

## A detailed procedure for genome sequencing analysis of phages

To sequence the genome of bacteriophage, the phage genomic DNA was extracted from different purified phage particles. First, to remove residual host DNA, 1 μg/mL DNase I and RNase A were

added to the concentrated phage solution for nucleic acid digestion overnight at 37°C. The digestion treatment was inactivated at 80°C for 15 min, followed by extraction with a Viral DNA Kit (Omega Bio-tek, USA) according to the manufacturer's instructions. Then, the genome sequencing was performed by Biozeron Biological Technology Co. Ltd (Shanghai, China). The detailed process of library construction, sequencing, genome assembly, and annotation was described below: (1) Library construction and Illumina HiSeq sequencing. Briefly, for Illumina pair-end sequencing of each phage, 0.2 µg genomic DNA was used for the sequencing library construction. Paired-end libraries with insert sizes of ~400 bp were prepared following the standard procedure. The purified genomic DNA was sheared into smaller fragments with a desired size by Covaris, and blunt ends were generated using the T4 DNA polymerase. The desired fragments were purified through gel-electrophoresis, then enriched and amplified by PCR. The index tag was introduced into the adapter at the PCR stage and we performed a library quality test. Finally, the qualified Illumina pair-end library was used for Illumina NovaSeq 6000 sequencing (150 bp*2, Shanghai Biozeron Co., Ltd). (2) Genome assembly. The raw paired-end reads were trimmed and quality controlled by the Trimmomatic (version 0.36) (*Pollet et al., 2011*) with parameters (SLIDINGWINDOW: 4:15, MINLEN: 75). Clean data were obtained and used for further analysis. We have used the ABySS software (http://www.bcgsc.ca/platform/bioinfo/software/abyss) to perform genome assembly with multiple-Kmer parameters and got the optimal results. The GapCloser software was subsequently applied to fill up the remaining local inner gaps and correct the single base polymorphism for the final assembly results. (3) Genome annotation. For bacteriophages, these obtained genome sequences were subsequently annotated by searching these predicted genes against non-redundant (NR in NCBI, 20180814), SwissProt (release-2021_03, http://uniprot.org) (*Dedysh and Ivanova, 2019*), KEGG (Release 94.0, http://www.genome.jp/kegg/) (*Buckley et al., 2006*), COG (update-2020_03, http://www.ncbi.nlm.nih.gov/COG) (*Brümmer et al., 2004*), and CAZy (update-2021_09, http://www.cazy.org/) (*Woebken et al., 2007*) databases.

## Growth assay of *P. stutzeri* 273 cultured in oligotrophic medium supplemented with different nitrogen sources and Phage-ZRK32

The assistance role of the bacteriophage induced from strain ZRK32 was tested in another deep-sea bacterium *P. stutzeri* 273 (*Wu et al., 2016*). Specifically, 50 µL freshly incubated *P. stutzeri* 273 cells were inoculated in 5 mL of either oligotrophic medium (10 g/L NaCl, 0.1 g/L yeast extract, 1 L sterilized distilled water), oligotrophic medium supplemented with 20 µL/L Phage-ZRK32 (without the extraction by chloroform), oligotrophic medium supplemented with 20 mM $NO_3^-$ or $NH_4^+$, oligotrophic medium supplemented with 20 mM of either $NO_3^-$ or $NH_4^+$, or 20 µL/L Phage-ZRK32. The cultures were then incubated under aerobic condition for 24 hr at 28°C. Three biological replicates for each culture condition were performed. The progress of the bacterial growth was monitored by measuring $OD_{600}$ values using a microplate reader (Infinite M1000 Pro, Switzerland) every 4 hr until cell growth reached a stationary phase.

## Acknowledgements

This work was funded by the Major Research Plan of the National Natural Science Foundation (Grant No. 92051107; 92351301), the NSFC Innovative Group Grant (No. 42221005), Science and Technology Innovation Project of Laoshan Laboratory (Grant No. LSKJ202203103; 2022QNLM030004-3), Shandong Provincial Natural Science Foundation (ZR2021ZD28), Strategic Priority Research Program of the Chinese Academy of Sciences (Grant No. XDA22050301), China Ocean Mineral Resources R&D Association Grant (Grant No. DY135-B2-14), Key Collaborative Research Program of the Alliance of International Science Organizations (Grant No. ANSO-CR-KP-2022–08), Key deployment projects of Center of Ocean Mega-Science of the Chinese Academy of Sciences (Grant No. COMS2020Q04), and the Taishan Scholars Program (Grant No. tstp20230637) for Chaomin Sun. We thank Dr. Diana Walsh from Life Science Editors for her great effort to improve the writing quality of our manuscript.

## Additional information

### Funding

| Funder | Grant reference number | Author |
|---|---|---|
| National Natural Science Foundation of China | 92051107 | Chaomin Sun |
| National Natural Science Foundation of China | 92351301 | Chaomin Sun |
| National Science Foundation of China | 42221005 | Chaomin Sun |
| Laoshan Laboratory | LSKJ202203103 | Chaomin Sun |
| Laoshan Laboratory | 2022QNLM030004-3 | Chaomin Sun |
| Shandong Provincial Natural Science Foundation | ZR2021ZD28 | Chaomin Sun |
| Chinese Academy of Sciences | XDA22050301 | Chaomin Sun |
| China Ocean Mineral Resources R&D Association | DY135-B2-14 | Chaomin Sun |
| Alliance of International Science Organizations | ANSO-CR-KP-2022–08 | Chaomin Sun |
| Center for Ocean Mega-Science, Chinese Academy of Sciences | COMS2020Q04 | Chaomin Sun |
| Natural Science Foundation of Shandong Province | tstp20230637 | Chaomin Sun |

The funders had no role in study design, data collection and interpretation, or the decision to submit the work for publication.

### Author contributions

Rikuan Zheng, Conceptualization, Validation, Investigation, Methodology, Writing - original draft; Chong Wang, Validation, Methodology, Writing - review and editing; Rui Liu, Investigation, Writing - review and editing; Ruining Cai, Methodology, Writing - review and editing; Chaomin Sun, Conceptualization, Supervision, Funding acquisition, Project administration, Writing - review and editing

### Author ORCIDs

Rikuan Zheng http://orcid.org/0009-0007-0275-0592
Chaomin Sun http://orcid.org/0000-0003-4802-184X

Reviewer #1 (Public Review): https://doi.org/10.7554/eLife.89874.3.sa1
Reviewer #2 (Public Review): https://doi.org/10.7554/eLife.89874.3.sa2
Author Response https://doi.org/10.7554/eLife.89874.3.sa3

## Additional files

### Supplementary files

• Supplementary file 1. Phenotypic and genotypic features of strain ZRK32 and the most closely related type strain *P. corsica* KS4$^T$.

• Supplementary file 2. The sugar utilization of strain ZRK32. +, Positive result or growth; −, negative result or no growth.

• Supplementary file 3. Marker genes used in the phylogenetic analysis. The DNGNGWU marker genes in phylosift refer to a suite of single-copy, protein-coding marker genes. All 37 DNGNGWU

marker genes were concatenated to construct maximum likelihood phylogenetic tree.
- Supplementary file 4. Primers used for real-time quantitative reverse transcription PCR (qRT-PCR).
- MDAR checklist

## Data availability

The full-length 16S rRNA gene sequence of strain ZRK32 has been deposited at GenBank under the accession number MW376756. The complete genome sequence of strain ZRK32 has been deposited at GenBank under the accession number CP066225. The raw sequencing reads from the transcriptomics analyses of ZRK32 strains cultured with different concentrations of yeast extract and nitrogen sources have been deposited into the NCBI Short Read Archive (accession numbers: PRJNA768630 and PRJNA694614, respectively). The genome sequence of Phage-ZRK32 has been deposited into the GenBank database with the accession number OP650935.

The following datasets were generated:

| Author(s) | Year | Dataset title | Dataset URL | Database and Identifier |
|---|---|---|---|---|
| Zheng R, Sun C | 2020 | Planctomycetota bacterium strain ZRK32 16S ribosomal RNA gene, partial sequence | https://www.ncbi.nlm.nih.gov/nuccore/MW376756 | NCBI Nucleotide, MW376756 |
| Zheng R, Sun C | 2020 | Planctomycetota bacterium strain ZRK32 chromosome, complete genome | https://www.ncbi.nlm.nih.gov/nuccore/CP066225 | NCBI Nucleotide, CP066225 |
| Zheng R, Sun C | 2021 | Transcriptome analysis of Planctomycetes strain ZRK32 | https://www.ncbi.nlm.nih.gov/bioproject/PRJNA768630 | NCBI BioProject, PRJNA768630 |
| Zheng R, Sun C | 2021 | Transcriptome analysis of ZRK32 | https://www.ncbi.nlm.nih.gov/bioproject/PRJNA694614 | NCBI BioProject, PRJNA694614 |
| Wang C | 2022 | Poriferisphaera phage ZRK32, partial genome | https://www.ncbi.nlm.nih.gov/nuccore/OP650935 | NCBI Nucleotide, OP650935 |

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
