## [Editor Report · eLife assessment]

This **important** study advances the understanding of physiological mechanisms in deep-sea Planctomycetes bacteria, revealing unique characteristics such as the only known Phycisphaerae using a budding mode of division, extensive involvement in nitrate assimilation, and release phage particles without cell death. The study uses **convincing** evidence based on experiments using growth assays, phylogenetics, transcriptomics, and gene expression data. The work will be of interest to bacteriologists and microbiologists in general.

---

## [Referee Report · Reviewer #1 (Public Review)]

The authors isolated a novel marine Planctomycetes bacterium with unique characteristics using a budding mode of division from the deep-sea cold seep sediment and named it Poriferisphaera heterotrophicis ZRK32. This work demonstrated that strain ZRK32 preferred nutrient-rich medium, moreover, the addition of nitrate or ammonia promoted the growth of strain ZRK32 and further caused the release of bacteriophage without killing the host. These results are interesting, well presented and documented in the revised manuscript.

---

## [Referee Report · Reviewer #2 (Public Review)]

Summary:

Planctomycetes encompass a group of bacteria with unique biological traits, the compartmentalized cells make them appear to be organisms in between prokaryotes and eukaryotes. However, only few of the Planctomycetes bacteria are cultured thus far, and hampers insight into the biological traits of this evolutionary important organisms.

This work reports the methodology details of how to isolate the deep-sea bacteria that could be recalcitrant to laboratory cultivation, and further reveals the distinct characteristics of the new species of a deep-sea Planctomycetes bacterium, such as the chronic phage release without breaking the host and promote the host and related bacteria in nitrogen utilization. Therefore, the finding of this work is of importance in extending our knowledge on bacteria.

Strengths:

Through combination of microscopic, physiological, genomics and molecular biological approaches, this reports isolation and comprehensively investigation of the first anaerobic representative of the deep-sea Planctomycetes bacterium, in particular in that of the budding division, and release phage without lysis the cells. Most of results and conclusions are supported by the experimental evidences.

---

## [Author Response]

The following is the authors’ response to the original reviews.

**eLife assessment:**
This important study advances the understanding of physiological mechanisms in deep-sea Planctomycetes bacteria, revealing unique characteristics such as the only known Phycisphaerae using a budding mode of division, extensive involvement in nitrate assimilation and release phage particles without cell death. The study uses convincing evidence, based on experiments using growth assays, phylogenetics, transcriptomics, and gene expression data. The work will be of interest to bacteriologists and microbiologists in general.

Response: Thanks for the Editor’s and Reviewers’ positive comments, which help us improve the quality of our manuscript entitled “Physiological and metabolic insights into the first cultured anaerobic representative of deep-sea Planctomycetes bacteria” (paper#eLife-RP-RA-2023-89874). The comments are all valuable, and we have studied the comments carefully and have made corresponding revisions according to the suggestions. Revised portions are marked in blue in the modified manuscript.

Please find the detailed responses as following.

**Public Reviews:**

**Reviewer #1 (Public Review):**
The authors of the manuscript cultivated a Planctomycetes strain affiliated with Phycisphaerae. The strain was one of the few Planctomycetes from deep-sea environments and demonstrated several unique characteristics, such as being the only known Phycisphaerae using a budding mode of division, extensive involvement in nitrate assimilation, and being able to release phage particles without cell death. The manuscript is generally well-written. However, a few issues need to be more clearly addressed, especially regarding the identification and characterization of the phage.

Response: Thanks for your positive comments. Please find the detailed responses as following.

**Reviewer #1 (Recommendations For The Authors):**
Line 75-77, add a reference for this statement.

Response: Thanks for your suggestion. We have added a reference (Fuerst and Sagulenko, 2011) for this statement in the revised manuscript (Line 77).

References related to this response:

Fuerst, J.A., and Sagulenko, E. Beyond the bacterium: planctomycetes challenge our concepts of microbial structure and function. Nat Rev Microbiol. 2011;9:403-413.

Line 124-134, add key statistics (such as ANI) of strain ZRK32 and KS4 to this section.

Response: Thanks for your suggestion. We added the key statistics of strain ZRK32 and KS4, and described as “Based on the 16S rRNA sequence of strain ZRK32, a sequence similarity calculation using the NCBI server indicated that the closest relatives of strain ZRK32 were Poriferisphaera corsica KS4T (98.06%), Algisphaera agarilytica 06SJR6-2T (88.04%), Phycisphaera mikurensis NBRC 102666T (85.28%), and Tepidisphaera mucosa 2842T (82.94%). Recently, the taxonomic threshold for species based on 16S rRNA gene sequence identity value was 98.65% (Kim et al., 2014). Based on these criteria, we proposed that strain ZRK32 might be a novel representative of the genus Poriferisphaera. In addition, to clarify the phylogenetic position of strain ZRK32, the genome relatedness values were calculated by the average nucleotide identity (ANI), the tetranucleotide signatures (Tetra), and in silico DNA-DNA similarity (isDDH), against the genomes of strains ZRK32 and KS4. The ANIb, ANIm, Tetra, and isDDH values were 72.89%, 85.34%, 0.97385, and 20.90%, respectively (Table S1). These results together demonstrated the strain ZRK32 genome to be obviously below established ‘cut-off’ values (ANIb: 95%, ANIm: 95%, Tetra: 0.99, isDDH: 70%) for defining bacterial species, suggesting strain ZRK32 represents a novel strain within the genus Poriferisphaera.” in the revised manuscript (Lines 124-139).

Fig. 2A missing description for figure key.

Response: Thanks for your comments. We modified the Figure 2A, shown as below:

**Author response image 1. sa3fig1:** Figure 2. Growth assay and transcriptomic analysis of P. heterotrophicis ZRK32 strains cultivated in basal medium and rich medium.

Regarding the page released, could this be a membrane vesicle-engulfed phage? I would recommend checking "Spontaneous Prophage Induction Contributes to the Production of Membrane Vesicles by the Gram-Positive Bacterium Lacticaseibacillus casei BL23" and "Chronic Release of Tailless Phage Particles from Lactococcus lactis" for further references.

Response: Thanks for your valuable comments. We carefully read these two papers and found that phage ZRK32 is most likely a membrane vesicle-engulfed phage. We added the corresponding description as “Moreover, it has recently been reported that the tailless Caudoviricetes phage particles are enclosed in lipid membrane and are released from the host cells by a nonlytic mechanism (Liu et al., 2022), and the prophage induction contributes to the production of membrane vesicles by Lacticaseibacillus casei BL23 during cell growth (da Silva Barreira et al., 2022). Considering that strain ZRK32 has a large number of membrane vesicles during cell growth (Figure S9), we speculated that Phage-ZRK32 might be a membrane vesicle-engulfed phage and its release should be related to membrane vesicles.” in the revised manuscript (Lines 381-388).

References related to this response:

Liu Y, Alexeeva S, Bachmann H, Guerra Martníez J.A, Yeremenko N, Abee T et al. Chronic release of tailless phage particles from Lactococcus lactis. Appl Environ Microbiol. 2022; 88: e0148321.

Silva Barreira, D., Lapaquette, P., Novion Ducassou, J., Couté, Y., Guzzo, J., and Rieu, A. Spontaneous prophage induction contributes to the production of membrane vesicles by the gram-positive bacterium Lacticaseibacillus casei BL23. mBio. 2022;13:e0237522.

How were the reference sequences for Fig. S10-S13 retrieved, was it by blasting the phage gene against the entire NCBI database, or only the virus sequence within the NCBI? Please clarify this.

Response: Thanks for your comments. The reference sequences for Fig. S10-S13 were retrieved by blasting the phage gene against the entire NCBI database. We clarified this as “The reference sequences of four AMGs encoding amidoligase, glutamine amidotransferase, gamma-glutamylcyclotransferase, and glutathione synthase were retrieved by blasting the phage gene against the entire NCBI database, respectively.” in the revised manuscript (Lines 444-447).

**Reviewer #2 (Public Review):**
Summary:Planctomycetes encompass a group of bacteria with unique biological traits, the compartmentalized cells make them appear to be organisms in between prokaryotes and eukaryotes. However, only a few of the Planctomycetes bacteria are cultured thus far, and this hampers insight into the biological traits of these evolutionarily important organisms. This work reports the methodology details of how to isolate the deep-sea bacteria that could be recalcitrant to laboratory cultivation, and further reveals the distinct characteristics of the new species of a deep-sea Planctomycetes bacterium, such as the chronic phage release without breaking the host and promote the host and related bacteria in nitrogen utilization. Therefore, the finding of this work is of importance in extending our knowledge of bacteria.

Response: Thanks for your positive comments.

Strengths:Through the combination of microscopic, physiological, genomics, and molecular biological approaches, this reports the isolation and comprehensive investigation of the first anaerobic representative of the deep-sea Planctomycetes bacterium, in particular in that of the budding division, and release phage without lysis of the cells.Most of the results and conclusions are supported by the experimental evidence.

Response: Thanks for your positive comments.

Weaknesses:1. While EMP glycolysis is predicted to be involved in energy conservation, no experimental evidence indicated any sugar utilization by the bacterium.

Response: Thanks for your comments. We have previously tested the sugar utilization of strain ZRK32, and now added this description as “Consistent with the presence of EMP glycolysis pathway in strain ZRK32, we found that it could use a variety of sugars including glucose, maltose, fructose, isomaltose, galactose, D-mannose, and rhamnose (Table S2).” in the revised manuscript (Lines 281-284).

1. "anaerobic representative" is indicated in the Title, the contrary, TCA in energy metabolism is predicted by the bacterium.

Response: Thanks for your valuable comments. Currently, anaerobic microorganisms can use other alternative electron acceptors (such as sulfate reducers, nitrate reducers, iron reducers, etc) in place of oxygen for the TCA cycle. For example, Proteus mirabilis uses the whole oxidative TCA cycle without using oxygen as the final electron acceptor when it performs multicellular swarming (Alteri et al., 2012). In this study, all the genes involved in the TCA cycle were present in anaerobic strain ZRK32 and most of them are upregulated, thus we speculate that it might function through the complete TCA metabolic pathway to obtain energy. We added the related description as “Notably, when growing in the rich medium, the expressions of most genes involved in the TCA cycle and EMP glycolysis pathway in strain ZRK32 were upregulated (Figure 2B-D, Figure S5B and Figure S6), suggesting that strain ZRK32 might function through the complete TCA metabolic pathway and EMP glycolysis pathway to obtain energy for growth (Figure S8) (Zheng et al., 2021b). Consistent with the presence of EMP glycolysis pathway in strain ZRK32, we found that it could use a variety of sugars including glucose, maltose, fructose, isomaltose, galactose, D-mannose, and rhamnose (Table S2). As for the presence of TCA cycle in the anaerobic strain ZRK32, we propose that it might use other alternative electron acceptors (such as sulfate reducers, nitrate reducers, iron reducers, etc) in place of oxygen for the TCA cycle, as shown in other anaerobic bacteria (Alteri et al., 2012).” in the revised manuscript (Lines 277-287).

References related to this response:

Alteri CJ, Himpsl SD, Engstrom MD, Mobley HL. Anaerobic respiration using a complete oxidative TCA cycle drives multicellular swarming in Proteus mirabilis. mBio. 2012; 3(6): e00365-12.

1. The possible mechanisms of the chronic phage release without breaking the host are not discussed.

Response: Thanks for your valuable comments. The possible mechanism of the chronic phage release without breaking the host might be that it was enclosed in lipid membrane and released from the host cells by a nonlytic mechanism. We added the corresponding description as “Moreover, it has recently been reported that the tailless Caudoviricetes phage particles are enclosed in lipid membrane and are released from the host cells by a nonlytic mechanism (Liu et al., 2022), and the prophage induction contributes to the production of membrane vesicles by Lacticaseibacillus casei BL23 during cell growth (da Silva Barreira et al., 2022). Considering that strain ZRK32 has a large number of membrane vesicles during cell growth (Figure S9), we speculated that Phage-ZRK32 might be a membrane vesicle-engulfed phage and its release should be related to membrane vesicles.” in the revised manuscript (Lines 381-388).

References related to this response:

Liu Y, Alexeeva S, Bachmann H, Guerra Martníez J.A, Yeremenko N, Abee T et al. Chronic release of tailless phage particles from Lactococcus lactis. Appl Environ Microbiol. 2022; 88:e0148321. da Silva Barreira, D., Lapaquette, P., Novion Ducassou, J., Couté,Y., Guzzo, J., and Rieu, A. Spontaneous prophage induction contributes to the production of membrane vesicles by the gram-positive bacterium Lacticaseibacillus casei BL23. mBio. 2022;13:e0237522.

**Reviewer #2 (Recommendations For The Authors):**
Have you tested whether strain ZRK32 uses any sugars? If not, why it uses EMP pathway to obtain energy?

Response: Thanks for your comments. We have previously tested the sugar utilization of strain ZRK32, and now added this description as “Consistent with the presence of EMP glycolysis pathway in strain ZRK32, we found that it could use a variety of sugars including glucose, maltose, fructose, isomaltose, galactose, D-mannose, and rhamnose (Table S2).” in the revised manuscript (Lines 281-284).

Further discussion on possible mechanisms of the chronic phage release without breaking the host is expected.

Response: Thanks for your valuable comments. The possible mechanism of the chronic phage release without breaking the host might be that it was enclosed in lipid membrane and released from the host cells by a nonlytic mechanism. We added the corresponding description as “Moreover, it has recently been reported that the tailless Caudoviricetes phage particles are enclosed in lipid membrane and are released from the host cells by a nonlytic mechanism (Liu et al., 2022), and the prophage induction contributes to the production of membrane vesicles by Lacticaseibacillus casei BL23 during cell growth (da Silva Barreira et al., 2022). Considering that strain ZRK32 has a large number of membrane vesicles during cell growth (Figure S9), we speculated that Phage-ZRK32 might be a membrane vesicle-engulfed phage and its release should be related to membrane vesicles.” in the revised manuscript (Lines 381-388).

References related to this response:

Liu Y, Alexeeva S, Bachmann H, Guerra Martníez J.A, Yeremenko N, Abee T et al. Chronic release of tailless phage particles from Lactococcus lactis. Appl Environ Microbiol. 2022; 88: e0148321.

da Silva Barreira, D., Lapaquette, P., Novion Ducassou, J., Couté, Y., Guzzo, J., and Rieu, A. Spontaneous prophage induction contributes to the production of membrane vesicles by the gram-positive bacterium Lacticaseibacillus casei BL23. mBio. 2022;13:e0237522.

It is recommended that the writing is improved, including presentation style and grammar.

Response: Thanks for your comments. We have invited an English native speaker (Dr. Diana Walsh from Life Science Editors, USA) to revise our manuscript, which we hope to meet your approval.